# Developing a competency assessment framework for pharmacists in primary health care settings in India

**Sanjeev Kumar[1]\*, Purnima Bhoi[2], Manjiri Sandeep Gharat[3,4,5], Guru Prasad Mohanta[6]**

1 Health Systems Transformation Platform, New Delhi, India, 2 KIIT School of Public Health, Bhubaneswar, India, 3 Prin. K.M. Kundnani Pharmacy Polytechnic, Ulhasnagar, India, 4 International Pharmaceutical Federation (FIP), The Netherlands, 5 Community Pharmacy Division-The Indian Pharmaceutical Association (CPA-IPA), Mumbai, India, 6 Department of Pharmacy, Annamalai University, Chidambaram, India

\* snjvkumar386@gmail.com

## Abstract

### Background

Competency frameworks in the pharmacy profession define the necessary practice standards and establish benchmarks for work accountability and career progression. Pharmacists are integral to primary healthcare, and assessing their competencies is essential for improving the performance of primary healthcare services. Although several countries have developed competency frameworks for pharmacists in primary healthcare, such frameworks are currently lacking in India.

### Methods

This study aimed to develop a competency assessment framework for in-service pharmacists in Indian public primary healthcare settings. For which, a five-stage consultative process was followed. In the first stage, the systematic literature review was conducted to identify pharmacist competencies in the primary healthcare setting. After that, an expert consultation was organized to develop consensus among experts on competencies and its behaviours. Competency assessment tools were then developed based on the literature and later finalized through experts' agreements during consultation. Finally, the tools were tested in one of the public primary healthcare facilities.

### Results

In stage one, the systematic literature review identified 20 competencies and 175 associated behaviors distributed across four domains: Pharmaceutical Health, Pharmaceutical Care, Organization and Management, and Professional/Personal. Expert consultations resulted in the consensus on 11 roles as domains of pharmacists in Indian primary healthcare settings during stage 2. During the expert consultation, each one of 11 pharmacist's role and competencies and behaviours was discussed and consensus was arrived on 26 competencies and 107 behaviours. For which, under the role of pharmacy management,

**Data availability statement:** All relevant data are within the paper and its Supporting Information files.

**Funding:** The author(s) received no specific funding for this work.

**Competing interests:** Authors with competing interests.

**Abbreviations:** LMIC, Low- and middle-Income Countries; PHC, Primary health care; WHO, World Health Organization; PSA, Pharmaceutical Society of Australia; AFPC, Association of Faculties of Pharmacy; CAT, Competency Assessment Tool; CBT, Competency-based Training; FIP, International Pharmaceutical Federation; NSQ, Not of Standard Quality; HSTP, Health Systems Transformation Platform; CPD-IPA, Community Pharmacy Division, Indian Pharmaceutical Association.

one of the critical competency for primary health care pharmacist is the knowledge about the layout and infrastructure of the pharmacy/store and how to adapt it to optimize pharmacy services. For this competency, one of the behaviour is assessment of the existing infrastructure and identifying gaps and opportunities for improvement.

After that in stage 3, a competency assessment tool was developed, and consensus was made on it in stage 4 during expert consultation. The developed tool contains instruments such as a questionnaire to assess knowledge and attitude, and observational checklist, mini-clinical exercises for specific conditions, and simulation exercises to assess skills.

## Conclusions

This study successfully developed a competency assessment framework for in-service pharmacists in Indian public primary healthcare settings. The framework encompasses 24 competencies and associated behaviors, covering 11 roles of pharmacists in Indian primary healthcare settings. For example, under the pharmacy management role, some of the competencies include planning and procurement, inventory management, storage conditions, digital literacy to handle supply chain-led IT Systems, NSQ Medical Products Management, etc.

The developed competency assessment tool assessing knowledge, attitude and skills provides a comprehensive framework for assessing pharmacist competencies and identifying competency gaps. The framework can be used to capacitate pharmacists and improve their performance. It improves their performance in primary healthcare settings, and enhance the delivery of healthcare services in India. Additionally, it fills a critical gap in the existing literature and can serve as a valuable resource for policymakers, educators, and healthcare professionals involved in pharmacy practice in primary care settings.

## Introduction

### Importance of pharmacist competency in primary health care setting

Pharmacists play an important role in primary health care [1]. In primary health care settings, Pharmacists as a team members engage in direct patient care and medication management services (MMS) for ambulatory patients, development of long-term relationships, coordination of care, patient advocacy, wellness and health promotion, triage and referral, and patient education and self-management [2].

Globally, including in low- and middle-income countries (LMICs), the role of pharmacists in primary health care is becoming vital because they support primary health care in meeting the elevated burden of chronic care [1]. The Primary Health Care Performance Initiative (PHCPI) framework has identified providers' competence as a critical driver to improve PHC performance [3]. In economic terms, an Australian study on the economic impact of increased clinical intervention rates in community pharmacies found that adequately trained and remunerated pharmacists generated savings (on health care, medicines, and pharmacy practice costs) six times greater than those of a control group with no access to the same education or remuneration. It was estimated that adequately trained and remunerated pharmacists would save the health care system 15 million Australian dollars (approx. US$100 million) a year [4]. Similar findings are reported from the USA [5]. Hence, the role of competent pharmacists in PHC performance is crucial.

## Pharmacists

The World Health Organization (WHO) defines pharmacists as "*healthcare professionals whose professional responsibilities and accountabilities include seeking to ensure that people derive maximum therapeutic benefit from their treatments with medicines*" [6]. A pharmacist usually works in one of the three settings, i) Pharmaceutical Industry, ii) Practice Setting, and iii) Others. Other settings include academics, regulators (government), and clinical research. In a practice setting, there are three areas: Community Pharmacy, Hospital Pharmacy, and Clinical Pharmacy [7]. The Community pharmacist is defined as "an individual currently registered and who works according to legal and ethical guidelines to ensure the correct and safe supply of medical products to the public" [8]. While, hospital pharmacists are defined as a "individual currently registered and who works in a hospital pharmacy service, primarily within the public/private sector and responsible for ensuring the safe, appropriate and cost-effective use of medicines" [8]. In India, Primary health care is being delivered through the Primary health centres, where the hospital pharmacists are in place and contributing in maintaining and improve people's health by advising and information and supplying prescription medicines. Pharmacists are well established in providing primary health care services in facilities. However, their role and contribution to primary health care teams, including roles in community pharmacy practice, are health care reforms in many countries [9].

In the growth of pharmacy practice in India, the health system is encountering multi-pronged challenges, including a shortage of trained teachers, limited skills demonstration facilities for teaching institutions, poor networking among the teaching institutions, limited pharmacist involvement with the community, limited interaction between clinician and pharmacist in patient management [10]. These challenges hinder pharmacists from acquiring competency and further developing competence in pharmacy practice.

## Competence & competency, competency assessment framework and competency assessment tool

Competence is one of the critical determinants of improving health outcomes and Client Satisfaction [11]. Provider's competence is the state of proficiency to perform the required practice activities to the defined standard, which includes the requisite competencies to successfully and effectively deliver high-quality services [11,12]. Providers or individuals develop competence through education, training, and work experience [13]. However, competency is the ability of a person to integrate knowledge, skills, and attitudes in their performance of the tasks in a given context [12]. Competence is dynamic and multidimensional, changing with time, experience, and context/ setting, while competencies are durable, trainable, and measurable through the expression of behaviors. The World Health Organization defines a competency framework "as an organized and structured representation of a set of interrelated and purposeful competencies" [12].

## Status of existing pharmacist competency assessment in primary health care settings

In Australia, professional bodies such as the royal pharmaceutical society (RPS) and the Pharmaceutical Society of Australia (PSA) have developed role descriptions and mapped the competencies for General Practice pharmacists [14]. In Canada, the Association of Faculties of Pharmacy (AFPC) educational outcomes for Canadian pharmacists have illustrated the competencies expected of pharmacists in the primary health care setting, either in the community or the primary health care team [9]. Similarly, In the United States of America, the role and competencies of pharmacist working in Primary Care has also been listed [2]. The

United Kingdom has also invested in integrating the clinical pharmacist into general practice since 2019, primarily to meet the workforce shortage [15]. Similarly, In the Netherlands, the Clinical Pharmacist has already been part of General practice [16]. Saudi Arabia has identified the competencies of Primary Care pharmacists for adults and paediatrics [17].

Many countries, primarily the global North and a few from the global South have defined the roles and competencies of Pharmacists in Primary Health Care settings [18]. Apart from that, in 2010, The International Pharmaceutical Federation (FIP) also developed the comprehensive Global Competency Framework for the services provided by Pharmacy Workforce, which is widely used globally [13].

## Availability of pharmacists in primary health care in India

India National Health Profile, 2022 reports that the total number of registered pharmacists in India until 2021 was 1,334,198. Of these, 24135 pharmacists were employed in public primary health centers [19]. The study conducted in Andhra Pradesh recorded that Primary health centers are experiencing a shortage of 13.3% of pharmacists as per Indian Public health standards [20].

## Rationale- why do we need a pharmacist competency assessment framework for India

Healthcare need is increasing because of rising global chronic conditions. This has forced the health system to extend primary Care Services to make them more comprehensive to serve the population need [21]. Understanding the critical role of pharmacists, the UK, Canada, The Netherlands, and other countries have invested in pharmacists to build competencies to improve its performance and meet the primary healthcare needs.

However, in LMICs, it is yet to begin, including India. In India, the situation is further complicated by the double burden of infectious and non-communicable diseases and increasing anti-microbial resistance [22,23]. Due to this, under the comprehensive primary health care package, twelve services have been provisioned in public primary health care institutions. It has significantly increased the pharmacist challenges in delivering health services, which were already facing challenges in competencies due to pre-service education and in-Service training [24].

Identifying the pharmacist's competency gap is critical for its capacitation in which the Competency Assessment Framework plays a significant role by assessing its competencies. Currently, no pharmacist competency assessment framework exists for Indian settings that can be used to assess competency for in-service pharmacists in primary health care. Hence, developing a competency assessment framework will bridge this gap, which will further help capacitate Primary Health Care Pharmacists.

## Study objective

The objective of this study was "to develop the competency assessment framework to assess the In-Service Pharmacist competency in public Primary Health Care setting." The specific objective was to identify Pharmacists' competencies and develop Competency Assessment Tools for in-service pharmacists in public primary healthcare settings.

## Methods

### Study design

We followed a five-stage Fig 1 consultative process to develop the Pharmacist competency assessment framework. The stage-wise details are as below.

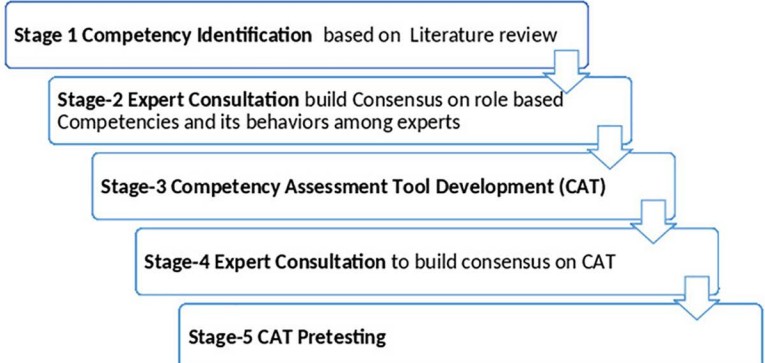

**Fig 1. Study Stages.**

## Stage 1 competency identification

In this phase, we did a systematic literature review Fig 2 to identify the Pharmacists' competencies in the primary health care setting to identify the scientific and grey literature between 2000 to 2020 on PubMed and SSRN. For PubMed, we used the search strategy as; ((("Primary Health Care/education" [Mesh] OR "Primary Health Care/standards"[Mesh])) AND "Pharmacists"[Mesh]) AND ("Professional Competence"[Mesh] OR "Clinical Competence"[Mesh] OR "Cultural Competency"[Mesh] OR "Social Skills"[Mesh]). Two extra filters were used, English and Humans. This strategy retrieved four papers. Similarly, for SSRN, "Pharmacist" as a title was searched, and then "Competence." This search retrieved only one paper. Google Scholar was also used to retrieve papers using two search strategies; "Primary Care Pharmacist Competency" and "Community Pharmacist Competency." Through this, 14 and seven papers were retrieved, respectively. Because the competencies have been minimally explored scientifically and are primarily available as guiding documents, Google has been used to search such literature [25]. Through this, 11 papers were retrieved. 38 papers were found from all the sources, of which five were duplicates. The remaining 33 articles were studied in detail, and later, twenty-five articles were removed. Finally, thirteen articles were selected that illustrated at least one domain of Pharmacist Competencies.

**Stage-2 Expert Consultation:** to build consensus on Competencies: In the second stage, we organized expert consultation to build consensus on Competencies and behaviours. This consultation was held virtually on 10th March 2022.

**Stage 3 Competency Assessment Tool Development:** In this stage, based on the literature, we developed the Instruments to assess the Competencies and their associated behaviors in terms of knowledge, attitude, and skills. These include a questionnaire, observation checklists, mini-clinical exercises and simulation exercises.

**Stage 4 Expert Consultation:** to build consensus on CAT: In this phase, we organized expert consultation on 14th November 2022 in Bhubaneswar, Odisha, to achieve consensus on Competency Assessment Tool instruments.

**Stage-5 Pretesting of Competency Assessment Tool:**–After converting the English version of the Competency Assessment Tool into a Bilingual language, i.e., English and Odia (Vernacular Language), we pretested it in One of the public primary health care setting facilities in Odisha, on 13th January 2023.

**Experts:** During two rounds of expert consultations, twenty experts participated. All the experts were trained Pharmacists and have been involved in delivering Pharmacy services in Primary Health Care Settings or as academia in Pharmacy education or involved in the

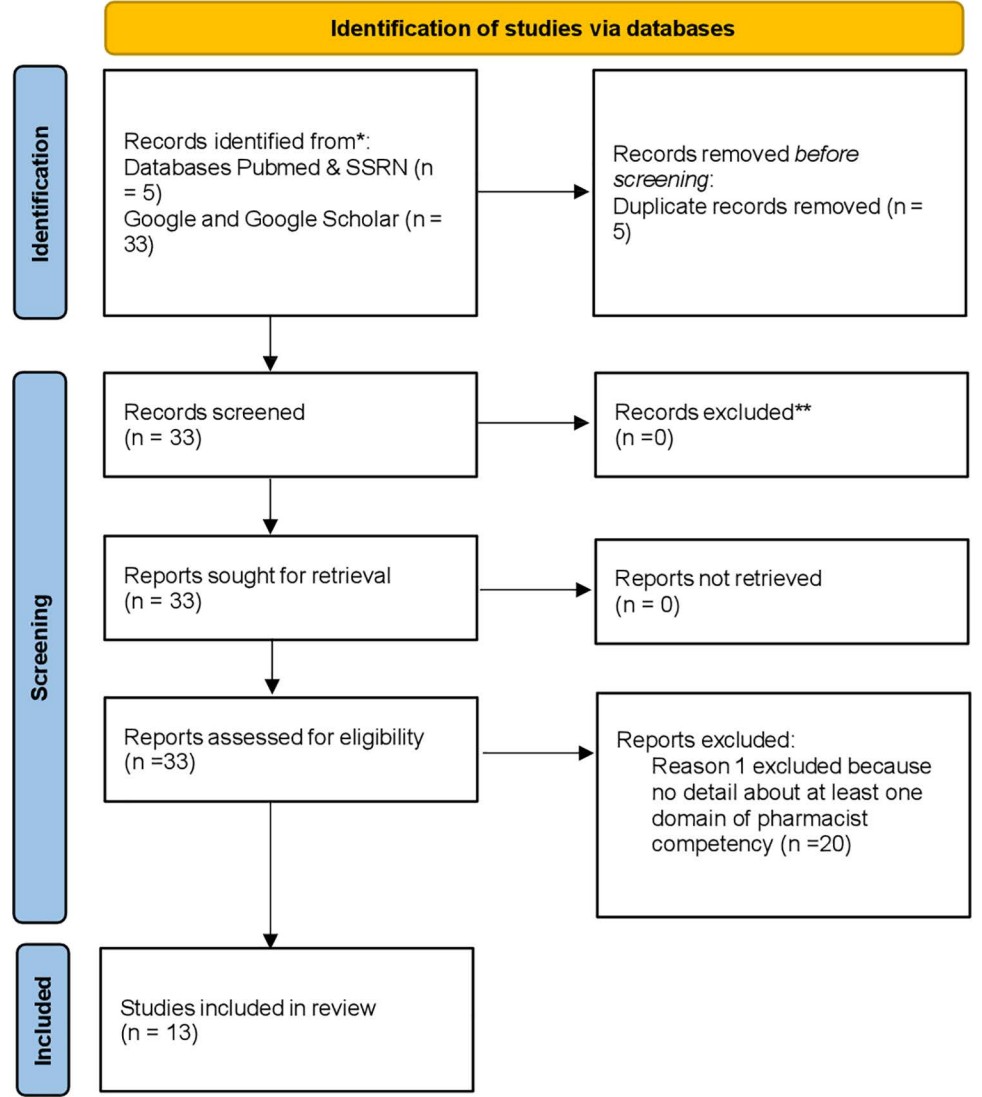

**Fig 2. PRISMA Selection of Studies.**

process of policy making as a multi-lateral organization representative or professional body representative or policy maker with over 20 Years of experience.

During the consultations, we obtained written consent from experts. Since we did not collect data from human participants, this study may be exempt from ethics committee approval.

## Results

### Stage -1 competencies identification

Through literature, we found 175 behaviours under 20 competencies, distributed among four domains like: Pharmaceutical Health, Pharmaceutical Care, Organization and Management, and Professional/personal. Under the pharmaceutical health domain, health promotion is one of the competencies. There are three behaviours; to assess primary healthcare needs; to advise on health promotion, disease prevention, and control, and a healthy lifestyle; and to provide

training programs for community staff and leaders on knowledge and skills for promoting health. Please refer to Table 1 for domains and their associated competencies and refer to S1 File for the detailed list of behaviours.

## Stage-2 expert consultation

Expert consultation was organized to build consensus among experts on pharmacist competencies and their associated behaviours. However, it emerged during the consultation that role plays a significant role in defining in-service competencies for any cadre of professionals. The consensus was made for eleven roles of Pharmacists in primary health care settings Table 2. After that, competencies and their associated behaviours were discussed, and consensus was achieved.

 **Role-1. Pharmacy Management:** In this role, during the consultation, we arrived at the Consensus on nine competencies and fifty-five behaviours, Table 3. In managing a Pharmacy, one of the critical Competencies is Knowledge about the Pharmacy, infrastructure, layout, and how to adapt it to optimize pharmacy services, including inventory management. Equally, knowing the drugs, consumables, and materials to be kept in store, their nature, use, and required storage conditions are important in managing pharmacy. Other competencies for Pharmacists in pharmacy management are maintenance of storage conditions, planning and procurement, inventory management, store verification, condemnation, and maintenance of registers and records. Knowledge about the Not of Standard Quality (NSQ) Medical Products and how to act appropriately is another vital competency for pharmacists in managing pharmacy. In this digital age, it is difficult for pharmacists to avoid the Competency around digital literacy, which helps maintain IT Based Supply chain systems like Drugs and Vaccine Distribution Management System (DVDMS) and e-Niramaya. Considering the limited

**Table 1. Domains and competencies.**

| Domains | Competencies | References |
|---|---|---|
| Pharmaceutical Health | Health Promotion | [13,26–31] |
| | Medicine Information and advice | [13,26,28–30,32] |
| Pharmaceutical Care | Assessment of Medicines | [13,27,28,30] |
| | Compounding Medicines | [13,26,29,30,32] |
| | Dispensing | [13,27,28,30] |
| | Medicines | [13,27,32] |
| | Monitor Medicines Therapy | [13,27,29,32] |
| | Patient Consultation and diagnostics | [13,27,28,30,32] |
| Organisation and Management | Budget and reimbursement | [13,30,33] |
| | Human Resource Management | [13,30,32] |
| | Service Improvement | [13,26,28,30] |
| | Procurement | [13,26] |
| | Supply Chain and management | [13,26,28,30] |
| | Workplace Management | [13,26–36] |
| Professional/ Personal | Communications Skills | [13,26–31,34–36] |
| | Continuing Professional Development | [13,28,30,31,34,35] |
| | Legal and Regulatory Practice | [13,26–28,33] |
| | Professional and ethical practice | [13,26–29,35] |
| | Quality Assurance and Research | [13, 26,28,29,35] |
| | Self-Management | [13,26–29,35,36] |

**Table 2. Pharmacist role.**

**Pharmacist Role in Indian Primary health care setting**

1. Pharmacy Management
2. Dispensing Medical Products
3. Ensure compliance with regulations on Medical Products
4. Assistance to Medical Officers in the preparation and implementation of different projects and programs
5. Outreach to the Community
6. Professional Practice
7. Ethical Practice
8. Communication
9. Workplace Management
10. Emergency Role
11. Continuing Professional Development.

**Table 3. Pharmacy management competencies.**

1. Knowledgeable about the Pharmacy layout, pharmacy infrastructure, and how to adapt it to optimize pharmacy services, including inventory management.
2. Knowledgeable about drugs, consumables, and materials to be kept in store, their nature, use, and storage conditions required.
3. Be able to maintain and ensure the required storage conditions.
4. Knowledgeable about the procedure for all types of planning and procurement (Pool Procurement)
5. Able to manage the inventory correctly.
6. Able to perform and fulfil the duties as a member of different committees in the institution or representing it.
7. Able to manage drugs and vaccine Distribution Management system (DVDMS)/eNiramaya, and other existing IT systems (Adequate digital literacy)
8. Able to conduct institutional store verification, condemnation, and maintenance of all registers and records relevant to the Pharmacy and pharmacy store.
9. Knowledgeable about the information and directions from authorities relating to banned and Not of Standard Quality (NSQ) medicines and acting appropriately

expertise about medical products at the primary health centre level, Pharmacists should have the Competency to attend different committees' meetings like Pharmacy (drug) and therapeutic committee, Annual Indent committee, Purchase committee, Condemnation committee, and Prescription audit committee.

**Role 2: Dispensing Medical Products:** During the consultation regarding this pharmacist's role in dispensing medical products, the consensus arrived at five competencies, Table 4, and sixteen behaviours, S2 File. One of the critical competencies is knowledge about drug details for dispensing, including generic/brand names, usage (Indication), dosage (Quantity and frequency), route of administration, how to use, and drug interaction/compatibility. However, Pharmacists also need to monitor the medical products' use and patient adherence, as appropriate, to ensure positive clinical outcomes. To improve patient adherence to medical products, Pharmacists need to encourage and facilitate. Hence, it is also identified as a competency. Another competency for pharmacists is recognizing the adverse drug reaction and supporting the primary health care team in managing the adverse reaction event.

**Role-3 Ensure compliance with regulations on Medical Products:** Medical products are a highly regulated commodity hence, dealing with them requires compliance with the applied regulations. Towards that, the consensus among the experts made during the consultation is that the Pharmacists in the primary health care setting should be able to guide the primary health care team of the health facility to ensure compliance with laws and regulations as applicable to the medical products, particularly medical products with the narcotics and psychotropic agents. Under this, the consensus was also made on five behaviours S2 File.

**Table 4. Dispensing medical products competencies.**

1. Knowledgeable about medical products details for dispensing, including generic/brand names, usage (Indication), dosage (Quantity and frequency), route of administration, how to use and drug interaction/compatibility,
2. Able to monitor medical product use and patient adherence, as appropriate, to ensure positive clinical outcomes.
3. Encourage and facilitate the patient for medication adherence.
4. Able to encourage use of medical products.
5. Able to recognize and support the Primary Health Care team in managing adverse drug reactions

**Role 4. Assistance to Medical Officer in preparation and implementation of different projects and Programs:** As a Pharmacist in a Primary Health Care setting, one role is to assist Medical Officer in preparing and implementing various health projects and programs, particularly in medical products management and its documentation. For details, please refer S2 File.

**Role 5. Outreach to the Community:** In this role, the consensus among the experts was made on two competencies and five behaviours, S2 File. One of them is to assist the primary health care team in organizing community outreach camps. During a disaster, pharmacists' roles become vital; another competency for Pharmacists is supporting the primary healthcare team in disaster management preparedness.

**Role-6 Professional Practice:** During the consultation, the experts unanimously agreed that Pharmacists should necessarily be knowledgeable about roles and responsibilities. Consensus was also reached on one behaviour S2 File.

**Role-7 Ethical Practice:** Ethical practice is central to delivering health services. Towards that, the consensus among the experts was made that the Pharmacist should be able to comply with ethical practices as a competency. The agreement was also made on three behaviours S2 File. One of the behaviour is the pharmacists need to be well versed with the of the Pharmacy Council of India Code of Ethics and as given in Pharmacy Practice Regulation.

**Role-8 Communication:** The Consensus among the experts during the consultation was that the Pharmacist should be able to communicate effectively in routine and emergencies. The Consensus among the experts was also made on the seven behaviours S2 File.

**Role-9 Workplace Management:** Workplace management is one of the typical roles that a pharmacist must play in a Primary Health Care Setting. For that, Consensus arrived during the consultation that one competency which every pharmacist should be able to manage the workplace. There were five behaviours on which the expert agreed. S2 File.

**Role-10 Emergency Role:** Under the emergency role, the Consensus among the experts during the consultation was made on three competencies. Pharmacists should essentially have the Competency to perform basic diagnostic tests/ Checks (Diagnostic tests include carrying out Point of Care tests like Blood pressure, Blood Glucose level using Glucometer, Pulse Oximetry, Peak flow tests, Weight, height checks and BMI Calculations, Snellen's Chart, Pulse, Respiration Rate, Hb using strips, Pregnancy testing, other spot tests using testing kits, etc.), support other health professionals, and handle medical devices/equipment. The government of Odisha, India, has permitted Pharmacists to provide treatment services for minor ailments without doctors [37]. Considering this state government initiative and the need, the experts arrived at the Consensus that pharmacists should also have the competencies to manage minor ailments by recommending medical products for diseases like `Fever, Malaria, Upper Respiratory Tract Infection, Skin diseases (Scabies, Ringworms), Helminthiasis, Acid Peptic Disease (APD), Diarrhoea, Minor Injury without Medico-Legal Cases, Superficial burns without Medico-Legal Cases (MLC), Drainage of abscess, etc. Regarding six behaviours in this role, experts arrived at consensus during the consultation.

**Role 11. Continuing Professional Development:** Health Services updates procedures and medical products based on global scientific advancements. Accordingly, the government and other stakeholders keep revising the guidelines. Understanding the volatility of different competencies, experts agreed that the Pharmacist should be willing to participate in Continuing Professional Education/ Development programs to update knowledge and practice. Experts also agreed on two behaviours for the said competency.

## Stage -3. Competency assessment tool (CAT) development

In this stage, to assess Pharmacist Competency based on literature, we developed Competency Assessment Tool. We developed the questionnaire to assess the Pharmacist's knowledge and attitude; under this, 256 questions were constructed, Table 5. The 143 questions were for knowledge, and the rest 113 were for attitude. Further, to assess the Pharmacist's skills, we used three assessment instruments, Observational Checklists, Mini Clinical Evaluations for Specific Conditions, and Simulation exercises [12,38]. Under Observation, we identified twenty-four skills; under mini-clinical evaluation exercises, we identified thirty skills and four skills under simulation exercises. For the Observation and Mini Clinical Evaluation of selected conditions, we identified the tracer elements for the skills and listed them as means of verification. To standardize the score based on means of verification, we developed the standard score definitions, Table 6 for observations and mini-clinical evaluation of selected conditions.

## Stage-4 expert consultation

In this stage, to develop the consensus among the experts on the Competency Assessment Tool, we organized an expert consultation on Nov 14, 2022, in Bhubaneswar. Consensus arrived on the Ninety-Six Questions during this consultation to assess knowledge and attitude. Out of ninety-six, forty-eight were for knowledge, while the rest were for attitude. On Observational skills, Consensus arrived at 20 skills, while it was thirty for Mini Clinical evaluation for specific conditions. For Simulation exercises, experts were in a consensus for all four skills, Table 5.

## Stage-5 CAT pre-testing

In this stage, the English version of CAT was converted into the bilingual format in English and Odia, S3 File. After that, we pre-tested in one of the public primary health care settings facilities in Khorda district, Odisha, India. We found that technical words like inventory management, not of Standard Quality, Supply Chain, prescription error, and prescription validation were comparatively more understandable in English than in Odia. We also understood that in place of the Drugs and Vaccines Distribution Management System (DVDMS), E-Niramaya, an IT-based supply chain system, is being used at the facility level. Accommodating these learnings, we developed the final version of bilingual CAT.

**Table 5. Competency assessment tool-instrument details.**

| Competency Elements | Instruments | Number of Questions/ Skills | |
|---|---|---|---|
| | | Pre-Consultation | Post Consultation |
| Knowledge | Questionnaire | 143 | 48 |
| Attitude | Questionnaire | 113 | 48 |
| Skills | Observational Checklist | 24 | 20 |
| | Mini Clinical Exercises for Specific Conditions | 30 | 30 |
| | Simulation Exercises | 4 | 4 |

**Table 6. Score definition.**

**Score Definition**
A. None—No demonstrated skills at all/does not perform the task(s) completely.
B. Limited Demonstrated very limited strengths/skills in this area.
C. Some—Demonstrated some ability/skills in this area.
D. Strong—Demonstrated strong skills/strength in this area.
E. Excellent—Demonstrated excellent skills/strength in this area.
F. Not applicable
G. Don't know- not even heard about that skill.
H. Skill limitation is related to resource limitations

## Discussion

We developed the competency assessment framework for In-Service pharmacists in primary healthcare settings. Under this, we identified the role-based competencies and their associated behaviours. Along with the competencies and their related behaviours, we developed the assessment tool to determine the degree of competency among in-Service pharmacists. In this way, this assessment framework will be helpful for policymakers and implementers in designing, monitoring, and evaluating pharmacist competencies. It will further improve pharmacist performance in the primary healthcare facilities.

**Factors Influencing Pharmacist Competency:** One of the factors that influence pharmacist competency the most is pre-service education. Teaching institutions engaged in pharmacist production face challenges in ensuring the availability of trained teachers. Qualified teachers who have exposure to be in the practice domain are very limited. Acquiring skills is not possible unless the teaching institutions have skills demonstration facilities. The shortage of skill demonstration facilities is also becoming a hindrance to developing skills during pre-service education and further prevents Pharmacists from entering pharmacy practice [10]. Another challenge of Pre-service education is the outdated curriculum and education regulations [39].

The National Education Policy 2020 emphasizes the importance of skill-based training in pharmacy education. Therefore, it is essential for the Pharmacy Council of India, the apex body responsible for regulating education and the profession in India, to update the pharmacy curriculum based on current health industry requirements [40]. In 2020, the curriculum for the diploma in pharmacy was revised for the first time in three decades. To prevent such lengthy delays in the future, the Pharmacy Council of India should establish a mechanism for periodically reviewing and updating curricula to align with industry and health sector needs.

The University Grants Commission (UGC) of the Government of India is tasked with ensuring quality in higher education, including pharmacy programs. One of its key responsibilities of UGC is to maintain teaching standards in higher education [41]. To address the impending shortage of skilled pharmacy educators, the UGC could design programs focused on enhancing the teaching competencies of pharmacy instructors. This initiative would undoubtedly help improve the overall quality of pharmacy education.

For the public health system, the Union and State governments have attempted to capacitate in-service Pharmacists except under the Revised National Tuberculosis Training Program [42]. Some States have made specific efforts to capacitate pharmacists, but even those efforts were limited to Store Management and Rational use of Drugs [43].

## Importance of competency assessment framework in developing competency-based training for pharmacists

The Competency assessment framework is the foundation for designing competency-based training (CBT) program for pharmacists, which helps them in improving their performance,

irrespective of the settings [44]. The framework provides structured tool to assess the competencies in terms of knowledge, attitude, and skills and helps to understand the role in which the pharmacist is experiencing competency gaps with specific knowledge, attitude, and skills. The framework also helps design competency-based training manuals and develop a pool of trainers to capacitate and bridge the competency gaps.

In India, there are lacunae in the design of the training program for capacitating in-service pharmacist [45]. Competency-based training (CBT), a structured training and assessment system that allows individuals to acquire skills and knowledge to perform work activities to a specified standard, is a globally accepted solution to capacitate the in-service professional [46]. For which the developed Competency assessment framework, based on the role-based competencies and their behaviour, along with the competency assessment tool, will play a foundational role in developing the Competency-based training manual and other resource materials for Pharmacists in primary health Care Settings.

## Strengths and limitations

The strength of this study is the consultative approach through which the Competency assessment framework has been developed. In this process, twenty experts in community pharmacy took part. These experts have over 20 Years of experience in teaching, policymaking, and practice from Teaching Institutions, Union and State Governments, professional bodies, Private retail pharmacy, and Public Primary Health Care facilities. In both consultations, twenty experts participated; hence, we recognize that the experts who participated in this study cannot be considered the complete representation of all the stakeholders for the In-Service Pharmacist Competency development initiative.

In method, In the first stage to identify literature-based competencies, we searched the literature from 2000 to 2020. We likely missed some of the relevant old and recent pieces of literature. Attitude assessment of a professional is a complex process; however, we developed the questionnaire to assess Pharmacists' attitudes. It is possible that the questionnaire to assess attitude cannot capture the responses entirely and accurately.

Apart from that, we did not use any tool to check the validity of the score definition for Skills assessment (Observation and Mini Clinical Exercises). We also understand that we pretested CAT in only one public facility in Odisha, India, and did not use any quantitative tool to check its validity.

The developed competency assessment framework is comprehensive. Hence, administering CAT to assess in-service pharmacist competency is time-consuming. As professionals' role evolves over time based on service requirements, this competency assessment framework will need periodic review and updating.

For the first time, this study has developed role-based competencies and competency assessment tools for in-service pharmacists in the Indian context; hence, if any new addition happens with the Pharmacist role, based on the extended role, the competencies, and their associated behaviours must be included accordingly.

## Conclusion

This study has developed a competency assessment framework for in-service primary healthcare pharmacists. It includes a set of competencies and a competency assessment tool designed to evaluate these competencies. The competencies and associated behaviors have been defined based on the roles of in-service pharmacists. The competency assessment tool evaluates pharmacists' knowledge, attitudes, and skills to assess their competencies.

The competency assessment framework will help identify the training needs of in-service pharmacists in primary healthcare settings. It will also assist the primary healthcare system in designing competency-based training programs. This type of training is expected to enhance pharmacists' competencies, ultimately improving the quality of services provided in primary healthcare facilities.

**Practice Implication:** Using this Competency Assessment framework, the health system implementers can identify the gap in serving Pharmacists' competencies in primary health care settings. Further, it can be instrumental in developing and designing competency-based training to upskill them.

## Supporting information

**S1 File. This is the S1 File literature based competencies and its behaviour List.** (PDF)

**S2 File. This is the S1 File pharmacists role based competencies and behaviours List.** (PDF)

**S3 File. This is the S1 File competency assessment tool.** (PDF)

## Acknowledgments

We would like to thank Mr. Rajeev Sadanandan, Dr. N. Devadasan, Mr. Sunil Nandraj, Dr. Kumaravel Ilangovan, Ms. Pallavi Gupta from HSTP for their useful suggestions on development of this paper. We also recognize the contributions of the experts who participated in the study consultations.

## Author contributions

**Conceptualization:** Sanjeev Kumar.

**Data curation:** Sanjeev Kumar, Purnima Bhoi, Manjiri Sandeep Gharat.

**Formal analysis:** Sanjeev Kumar.

**Investigation:** Sanjeev Kumar, Manjiri Sandeep Gharat.

**Methodology:** Sanjeev Kumar, Manjiri Sandeep Gharat, Guru Prasad Mohanta.

**Project administration:** Sanjeev Kumar, Purnima Bhoi.

**Supervision:** Guru Prasad Mohanta.

**Validation:** Sanjeev Kumar.

**Visualization:** Sanjeev Kumar.

**Writing – original draft:** Sanjeev Kumar.

**Writing – review & editing:** Sanjeev Kumar, Purnima Bhoi, Manjiri Sandeep Gharat, Guru Prasad Mohanta.

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
