## [Decision Letter · Decision Letter 0]

22 Oct 2024

PONE-D-24-33395Developing a competency assessment framework for pharmacists in primary health care settings in IndiaPLOS ONE

Dear Dr. Kumar,

Thank you for submitting your manuscript to PLOS ONE. After careful consideration, we feel that it has merit but does not fully meet PLOS ONE’s publication criteria as it currently stands. Therefore, we invite you to submit a revised version of the manuscript that addresses the points raised during the review process.

We look forward to receiving your revised manuscript.

Kind regards,

Naeem Mubarak, PhD

Academic Editor

PLOS ONE

Journal Requirements:

1. When submitting your revision, we need you to address these additional requirements. Please ensure that your manuscript meets PLOS ONE's style requirements, including those for file naming. The PLOS ONE style templates can be found at https://journals.plos.org/plosone/s/file?id=wjVg/PLOSOne_formatting_sample_main_body.pdf and https://journals.plos.org/plosone/s/file?id=ba62/PLOSOne_formatting_sample_title_authors_affiliations.pdf 2. Please provide additional details regarding participant consent. In the ethics statement in the Methods and online submission information, please ensure that you have specified (1) whether consent was informed and (2) what type you obtained (for instance, written or verbal, and if verbal, how it was documented and witnessed). If your study included minors, state whether you obtained consent from parents or guardians. 3. Thank you for stating the following in the Competing Interests section: "Authors with competing interests" Please confirm that this does not alter your adherence to all PLOS ONE policies on sharing data and materials, by including the following statement: "This does not alter our adherence to  PLOS ONE policies on sharing data and materials.” (as detailed online in our guide for authors http://journals.plos.org/plosone/s/competing-interests). If there are restrictions on sharing of data and/or materials, please state these. Please note that we cannot proceed with consideration of your article until this information has been declared.  Please include your updated Competing Interests statement in your cover letter; we will change the online submission form on your behalf. 4. We note that your Data Availability Statement is currently as follows: [All relevant data are within the manuscript and its Supporting Information files.] Please confirm at this time whether or not your submission contains all raw data required to replicate the results of your study. Authors must share the “minimal data set” for their submission. PLOS defines the minimal data set to consist of the data required to replicate all study findings reported in the article, as well as related metadata and methods (https://journals.plos.org/plosone/s/data-availability#loc-minimal-data-set-definition). For example, authors should submit the following data: - The values behind the means, standard deviations and other measures reported;- The values used to build graphs;- The points extracted from images for analysis. Authors do not need to submit their entire data set if only a portion of the data was used in the reported study. If your submission does not contain these data, please either upload them as Supporting Information files or deposit them to a stable, public repository and provide us with the relevant URLs, DOIs, or accession numbers. For a list of recommended repositories, please see https://journals.plos.org/plosone/s/recommended-repositories. If there are ethical or legal restrictions on sharing a de-identified data set, please explain them in detail (e.g., data contain potentially sensitive information, data are owned by a third-party organization, etc.) and who has imposed them (e.g., an ethics committee). Please also provide contact information for a data access committee, ethics committee, or other institutional body to which data requests may be sent. If data are owned by a third party, please indicate how others may request data access. 5. Please include captions for your Supporting Information files at the end of your manuscript, and update any in-text citations to match accordingly. Please see our Supporting Information guidelines for more information: http://journals.plos.org/plosone/s/supporting-information.

Additional Editor Comments:

The manuscript has strengths but needs major revisions to improve its quality and merit.

Reviewers' comments:

Reviewer's Responses to Questions

**Comments to the Author**

1. Is the manuscript technically sound, and do the data support the conclusions?

Reviewer #1: Yes

2. Has the statistical analysis been performed appropriately and rigorously? 

Reviewer #1: No

3. Have the authors made all data underlying the findings in their manuscript fully available?

Reviewer #1: Yes

4. Is the manuscript presented in an intelligible fashion and written in standard English?

Reviewer #1: Yes

5. Review Comments to the Author

Reviewer #1: Thank you, managing editorial, for giving me this opportunity to review the article “Developing a competency assessment framework for pharmacists in primary health care settings in India” in this prestigious journal. This article provides the initial primary stages and roles for developing the competency assessment framework for primary health care in India.

Abstract

In the background, it's crucial to highlight the significance of the competency framework for pharmacists, as it underpins their professional development and performance.

The method of assessment was insufficiently described.

To whom were the competency framework and associated behaviour discussed and agreed upon? Please clarify the statement.

In general, there is a lack of information about the competencies, just the number of competencies and the number of roles. Kindly add some lines in the result or conclusion about which competency framework for pharmacists has been introduced.

Introduction

Please add an example from the primary rather than the community set-up to elaborate on the importance of competency at the primary healthcare level.

Add a few lines about the availability of pharmacists at the primary healthcare level and then explain their importance to their country's prospects.

The authors are advised to specify the status of pharmacists at their country level because these studies are of Indian origin. So, the targeted approach will be more important than covering the whole world.

Please advise the authors to reference the statements, as no competency assessment has already been conducted in their country.

There is no introduction regarding the competency tool and globally available competencies for pharmacists at the primary level.

Method

Authors are advised to use the assessment tool for literature review for effective data collection and representation.

In stage two, on which basis the experts were selected for consultation.

In stage three, which instrument was used for assessing competencies? Please specify.

There should be numbered score definitions in the assessment tool.

Is there only English grammar pretesting required for the competency assessment tool?

Results

The competencies selected in the literature review are related to the in-service pharmacist or not. These competencies were related to pharmacists or taken as general.

On which basis were the eleven roles made for pharmacist competencies, and on which bases were they selected?

In role 2, how is it possible for a pharmacist to observe the optimal use of medical products?

In role 3, please specify which drugs require high compliance with regulations.

In role 4, although medical product management and documentation are solely the responsibility of the pharmacist as they are available in the pharmacy, the pharmacist assists the medical officer with these tasks.

In role 7, on which limit the ethical practice will be allowed. Please specified.

In role 10, Is it possible for the author to please specify the list of diagnostic tests and checks in emergencies?

In role 11, authors are advised to please specify the programs for continuing professional development in Indian origin.

Authors are advised to reduce the paragraph content as already described in the table.

Authors are advised to provide the details of the validation of the competency assessment tool.

Which tests were performed during the pre-testing of the competency assessment tool?

Discussion

The discussion regarding the factors related to coping with pre-service education is limited. The authors advised adding more possible measures to cope with these challenges at the government level.

The discussion regarding the importance and implications of competence assessment was very limited, making this portion weak.

6. PLOS authors have the option to publish the peer review history of their article (what does this mean? ). If published, this will include your full peer review and any attached files.

**Do you want your identity to be public for this peer review?** For information about this choice, including consent withdrawal, please see our Privacy Policy .

Reviewer #1: No

---

## [Author Response · Author response to Decision Letter 0]

6 Nov 2024

Sr. No. Section Comments Response Description-Response (Page and Line Number

1. Abstract In the background, it's crucial to highlight the significance of the competency framework for pharmacists, as it underpins their professional development and performance. Referring to the suggestion, I made the required changes. Please refer to ln No. 35 to 40 on page 2

2 The method of assessment was insufficiently described. Referring to the suggestion, I made the required changes. Please refer to ln No. 42 to 49 on page 2

3 To whom were the competency framework and associated behaviour discussed and agreed upon? Please clarify the statement. Referring to the suggestion, I made the required changes. Please refer to ln No. 53 to 54 on page 2

4 In general, there is a lack of information about the competencies, just the number of competencies and the number of roles. Kindly add some lines in the result or conclusion about which competency framework for pharmacists has been introduced. Referring to the suggestion, I made the required changes. Please refer to ln No. 88 to 101 on page 3.

5 Introduction Please add an example from the primary rather than the community set-up to elaborate on the importance of competency at the primary healthcare level. I tried my best to include articles from primary healthcare settings.

6 Add a few lines about the availability of pharmacists at the primary healthcare level and then explain their importance to their country's prospects. Referring the suggestion, made the necessary changes. Please refer to ln No. 198 to 202 on page 6.

7 The authors are advised to specify the status of pharmacists at their country level because these studies are of Indian origin. So, the targeted approach will be more important than covering the whole world. Yes, I tried my best to share status in an Indian setting Please refer to ln No. 211 to 217 on page 6.

8 Please advise the authors to reference the statements, as no competency assessment has already been conducted in their country We learned from literature search and expert consultation that there is no competency assessment framework.

9 There is no introduction regarding the competency tool and globally available competencies for pharmacists at the primary level. We discussed about the FIP competency framework and later in stage 1, we listed the competencies under different domains from various studies. Please refer to ln No. 179 to 197 on pages 5 and 6.

10 Method Authors are advised to use the assessment tool for literature review for effective data collection and representation. In stage 1, we used a systematic approach to searching literature through a defined search strategy and followed PRISMA guidelines. Please refer to Fig 1 for details.

11 In stage two, on which basis the experts were selected for consultation? We identified experts based on selection criteria such as education, years of experience, and current role. Please refer to the experts section. Please refer to ln No. 277 to 281 on page 8.

12 In stage three, which instrument was used for assessing competencies? Please specify. Referring to the suggestion, we made the changes. Please refer to ln No. 268 to 269 on page 8.

13 There should be numbered score definitions in the assessment tool. We developed the score definition for the instruments assessing skills as Observational checklists and mini-clinical exercises. Please refer to table 8 Score definition. Please refer to table 5 on page 14

14 Is there only English grammar pretesting required for the competency assessment tool? No, we pre-tested the competency assessment tool in bilingual format in English and Odia. Please refer to ln no 466 and 467.

15 Results The competencies selected in the literature review are related to the in-service pharmacist or not. These competencies were related to pharmacists or taken as general. Yes, the literature review is related to the pharmacists working in primary health care settings.

16 On which basis were the eleven roles made for pharmacist competencies, and on which bases were they selected? In stage 2, during the consultation, it emerged that rather than considering the domain, it is better to use the role as a basis to define competencies. These roles are primarily based on the Indian Public Health Standard, 2022, and later on, the consensus among experts, 11 roles were finalized. Please refer to ln no 318-320 on page 9.

17 In role 2, how is it possible for a pharmacist to observe the optimal use of medical products? In role 2, one of the competencies is- Able to encourage the use of medical products.

The pharmacist can encourage patients to return unused, unwanted, or expired medicines to the Pharmacy for safe disposal. Pharmacists can also maintain records of returned medical products to monitor usage further.

The additional file mentions these two ways as behaviors for the said competencies. For details, please refer to Additional file 2. Please refer to additional file 2.

18 In role 3, please specify which drugs require high compliance with regulations. Medical products with narcotics and psychotropic agents require high compliance with regulations.

We made the suggested changes. Please refer to ln no 362-363 on page 12.

19 In role 4, although medical product management and documentation are solely the responsibility of the pharmacist as they are available in the pharmacy, the pharmacist assists the medical officer with these tasks. Primary health Centres conduct various activities for public health programs, including Leprosy, Tuberculosis, and Vector disease control programs. In those cases, the Pharmacist is supposed to assist the medical officer with medical product management and documentation.

20 In role 7, on which limit the ethical practice will be allowed. Please specified. Regarding ethical practice, the consensus made among the experts as one of the behaviors is -

“Be well versed with the Code of Ethics of the Pharmacy Council of India and as given in Pharmacy Practice Regulation.” (Please refer to additional file 2 )

Made the required changes in the manuscript. Please refer to additional file 2.

21 In role 10, Is it possible for the author to please specify the list of diagnostic tests and checks in emergencies? We specified the list of diagnostics in the manuscript. These tests are described as behaviors in Additional File 2. Please refer to additional file 2

22 In role 11, authors are advised to please specify the programs for continuing professional development in Indian origin. In India, there is no structured program for pharmacists' continuing professional development. The National Health Mission, along with other initiatives, organizes capacity-building programs regularly. In light of this, one behavior that emerged from expert consensus is actively participating in the various training programs offered by the National Health Mission and other health programs established by the Ministry of Health and Family Welfare (MoHFW) and State Health Departments.

For details, please refer to additional file 2. Please refer to additional file 2.

23 Authors are advised to reduce the paragraph content as already described in the table. We removed two tables-table 4 and 5.

24 Authors are advised to provide the details of the validation of the competency assessment tool. Which tests were performed during the pre-testing of the competency assessment tool? We developed the CAT using a content validation (Stage 4) approach. Later, under stage 5, we pre-tested the CAT in one of the public health facilities to understand the construct regarding language clarity and local programmatic variations. Please refer to ln no 467 to 475 on pages 15 and 16.

25 Discussion The discussion regarding the factors related to coping with pre-service education is limited. The authors advised adding more possible measures to cope with these challenges at the government level. We made the needful changes. Please refer to ln no 493 to 515 on pages 16 and 17.

26 The discussion regarding the importance and implications of competence assessment was very limited, making this portion weak. We made the needful changes. Please refer to ln no 522 to 536 on page 17.

---

## [Decision Letter · Decision Letter 1]

16 Dec 2024

Developing a competency assessment framework for pharmacists in primary health care settings in India

PONE-D-24-33395R1

Dear Dr. Sanjeev Kumar,

We’re pleased to inform you that your manuscript has been judged scientifically suitable for publication and will be formally accepted for publication once it meets all outstanding technical requirements.

Kind regards,

Naeem Mubarak, PhD

Academic Editor

PLOS ONE

Additional Editor Comments (optional):

The manuscript requires no further changes.

Reviewers' comments:

Reviewer's Responses to Questions

**Comments to the Author**

1. If the authors have adequately addressed your comments raised in a previous round of review and you feel that this manuscript is now acceptable for publication, you may indicate that here to bypass the “Comments to the Author” section, enter your conflict of interest statement in the “Confidential to Editor” section, and submit your "Accept" recommendation.

Reviewer #2: All comments have been addressed

2. Is the manuscript technically sound, and do the data support the conclusions?

Reviewer #2: Yes

3. Has the statistical analysis been performed appropriately and rigorously? 

Reviewer #2: Yes

4. Have the authors made all data underlying the findings in their manuscript fully available?

Reviewer #2: Yes

5. Is the manuscript presented in an intelligible fashion and written in standard English?

Reviewer #2: Yes

6. Review Comments to the Author

Reviewer #2: The author has thoroughly addressed all the comments and suggestions. After reviewing the updated manuscript, I find the revisions satisfactory and in line with the objectives of the work. At this stage, I have no further suggestions or comments to add.

7. PLOS authors have the option to publish the peer review history of their article (what does this mean? ). If published, this will include your full peer review and any attached files.

**Do you want your identity to be public for this peer review?** For information about this choice, including consent withdrawal, please see our Privacy Policy .

Reviewer #2: No

---

## [Editor Report · Acceptance letter]

PONE-D-24-33395R1

PLOS ONE

Dear Dr. Kumar,

I'm pleased to inform you that your manuscript has been deemed suitable for publication in PLOS ONE. Congratulations! Your manuscript is now being handed over to our production team.

Kind regards,

on behalf of

Dr Naeem Mubarak

Academic Editor

PLOS ONE